# Evaluating the cost-effectiveness of polygenic risk score-stratified screening for abdominal aortic aneurysm

M. Kelemen[1,2], J. Danesh[1,2], E. Di Angelantonio[1,2], M. Inouye ®[1,2,3], J. O'Sullivan[4], L. Pennells ®[1,2], T. Roychowdhury[5], M. J. Sweeting[6], A. M. Wood ®[1,2], S. Harrison[7] & L. G. Kim ®[1,2] ✉

As the heritability of abdominal aortic aneurysm (AAA) is high and AAA partially shares genetic architecture with other cardiovascular diseases, genetic information could help inform AAA screening strategies. Exploiting pleiotropy and meta-analysing summary data from large studies, we construct a polygenic risk score (PRS) for AAA. Leveraging related traits improves PRS performance ($R^2$) by 22.7%, relative to using AAA alone. Compared with the low PRS tertile, intermediate and high tertiles have hazard ratios for AAA of 2.13 (95%CI 1.61, 2.82) and 3.70 (95%CI 2.86, 4.80) respectively, adjusted for clinical risk factors. Using simulation modelling, we compare PRS- and smoking-stratified screening with inviting men at age 65 and not inviting women (current UK strategy). In a futuristic scenario where genomic information is available, our modelling suggests inviting male current smokers with high PRS earlier than 65 and screening female smokers with high/intermediate PRS at 65 and 70 respectively, may improve cost-effectiveness.

Abdominal aortic aneurysm (AAA) is an enlargement of the abdominal aorta to greater than 3 cm diameter. To promote early detection and management of AAA (which reduces the likelihood of fatal aortic ruptures), authorities in the UK, Sweden, and Oslo, Norway, offer screening to all men aged 65[1]. By contrast, the US Preventive Services Task Force (USPSTF) recommends a one-off screening with ultrasound scan (USS) in men aged 65–75 who have ever smoked and recommends selective screening based on clinical judgement in men who have never smoked[2]. However, despite the introduction of such screening programmes, ruptured AAA continues to be a significant source of morbidity and mortality, exemplified by AAA's contribution to around 0.8% of deaths in men aged 65 and older and 0.4% in women aged 65 and older in England and Wales[3].

Rupture of AAAs occurs primarily in people not captured by current screening programmes. They include, for example women, men under 65, men invited to screening at 65 but who did not attend,

and men with a normal aortic ultrasound scan at age 65 but who develop an AAA later in life. A small number of ruptures also occur in screen-detected AAAs[4] either prior to reaching the threshold for elective intervention, whilst waiting for this intervention, or after being declared unfit for intervention. However, the prevalence of AAA is falling over time. For example, less than 1% of men screened as part of the UK National Health Service (NHS) abdominal aortic aneurysm screening programme (NAAASP) in 2021 were observed to have an AAA[3]. By contrast, the prevalence of AAA was 4.9% in the UK Multi-centre Aneurysm Screening Study (MASS)[5], carried out in 1997-9. This temporal decline in the population burden of AAA suggests that the cost-effectiveness of AAA screening may also be changing, encouraging consideration and evaluation of more targeted screening approaches[6]. The USPSTF has, for example, highlighted that research is needed to define the benefits of screening in particular subgroups, suggesting that a stratified approach may have net benefit over current

[1]British Heart Foundation Cardiovascular Epidemiology Unit, Department of Public Health and Primary Care, University of Cambridge, Cambridge, UK. [2]Victor Phillip Dahdaleh Heart and Lung Research Institute, University of Cambridge, Cambridge, UK. [3]Baker Heart & Diabetes Institute, Melbourne, Australia. [4]Division of Cardiology, Department of Medicine, Stanford University, Stanford, CA, USA. [5]Department of Genetics, Yale School of Medicine, New Haven, CT, USA. [6]Department of Population Health Sciences, University of Leicester, Leicester, UK. [7]Genomics PLC, Oxford, UK. ✉e-mail: lois.kim@medschl.cam.ac.uk

strategies[2]. One potential subgroup is current cigarette smokers, for whom a strong association with AAA[7] has been reported (7-fold increase in risk in men, and 15-fold in women).

As the heritability of abdominal aortic aneurysm (AAA) is high (perhaps as high as 70%[8]) and genome wide association studies (GWAS) have identified numerous variants associated with AAA susceptibility[9–11], there is interest in whether use of genetic information could supplement and/or support population screening strategies for AAA. Polygenic risk scores (PRS), which aggregate the effects of genetic variants across the genome, can help stratify populations to help identify individuals with higher risk of disease[12,13]. As AAA often co-occurs with other forms of cardiovascular diseases[14] and as the condition partially shares genetic architecture with other cardiovascular diseases[15], we hypothesise that methods[16–18] leveraging shared genetic effects across multiple correlated clinical traits should optimise the performance of an AAA PRS.

In this work, we develop a state-of-the-art PRS for AAA leveraging its pleiotropy with related traits. We then adapt a discrete event simulation model previously developed to evaluate the potential cost-effectiveness of screening females for AAA[19] to explore the potential clinical impact and cost-effectiveness of a stratified screening programme informed by our PRS.

## Results
### Overview of our study
Our analysis plan comprised three stages relating to (i) development of a novel PRS for AAA, (ii) evaluation of the association of this PRS with AAA, and (iii) use of a discrete event simulation model to assess the potential for using this PRS and smoking status to inform screening for AAA.

To develop our PRS, we assembled the largest to-date training dataset for AAA by combining information from multiple large biobanks (Methods). To further improve performance, based on the clinical experience of known co-morbidities of AAA we incorporated information from other cardiovascular diseases into our PRS, utilising a recently developed method that leverages shared genetic effects[16]. Our final PRS was built by LDpred2[20] and evaluated on a non-overlapping subset of participants in the UK Biobank study. We then tested our PRS's association with incident AAA using Cox regression models on the age time-scale. Finally, to explore how a PRS-informed age at invitation strategy may influence long-term clinical and cost outcomes, we deployed a previously validated discrete event simulation (DES) model for AAA screening, with PRS tertile-specific AAA prevalences estimated from a Fine and Gray regression model treating non-AAA mortality as a competing risk. Incremental net benefit estimates from the DES were then used to propose and evaluate policy recommendations. Full details of our methods can be found in the Online methods section. See Supplementary Fig. 1 for a summary of the study design.

### AAA polygenic risk score development
We benchmarked a number of different PRS and selected the best-performing model, which integrated information from all available AAA studies and also from GWAS summary data from two traits that shared genetic aetiology with AAA: coronary artery disease (CAD) and *AAA-related*. AAA-related is a composite phenotype of 21 conditions related to AAA (for example, Marfan's syndrome, myocardial infarction and hypertension, see Supplementary Table 2 for a full list of conditions). Leveraging these additional related traits provided a performance improvement in $R^2$ of 22.7% over just using information from AAA alone (0.00432 vs 0.00530). (). Our best-performing PRS was selected using area under the receiver-operator curve (AUC) in our test-set for prevalent/incident AAA risk combined (prevalent cases can be included here as only considering PRS in the model, thus eliminating the possibility of reverse causality). The full details of all PRS

models, their development and performance summaries can be found in the Methods section and Supplementary Table 3. The performance of the best PRS, evaluated using the incident time-to-AAA outcome in our test-set, is shown in Fig. 1.

### Association of PRS with AAA
Amongst the 91,731 individuals in the UK Biobank test set, 634 (1.7%) men and 106 (0.2%) women had an AAA event (i.e. events captured by the definition and data in UK Biobank, hereafter referred to as AAA; see Methods) during the follow-up period. 72,928 (79.5%) had complete data on all risk factors; of these, 464 men (1.5%) and 81 women (0.2%) had an AAA. Median follow-up was 12.0 (IQR 11.2 to 12.7) years. Only 222 (0.2%) individuals were censored. A summary of risk factors and missing data is provided in Supplementary Table 5; events and frequencies by sub-group are given in Supplementary Table 6.

Hazard ratios from multivariable Cox regression modelling are provided in Table 1, based on the complete case analysis. The results demonstrated a higher rate of AAA across the PRS risk groups - defined as tertiles of the PRS distribution - even after adjustment for the other risk factors. Compared to the low PRS risk group, the intermediate PRS risk group had a 2-fold higher hazard of AAA (HR 2.13, 95% CI 1.61 to 2.82), and the high PRS risk group a nearly 4-fold increase (HR 3.70, 95% CI 2.86 to 4.80). Alternatively, if PRS was modelled as a continuous predictor, the adjusted HR per 1 standard deviation increase was 1.77 (95% CI 1.63 to 1.93). As expected, the hazard of AAA was also higher in ex and current smokers compared to never smokers (HRs: 2.36, 95% CI 1.82, 3.05; and 7.74, 95% CI 5.83, 10.29, respectively). There was no evidence of an interaction between sex and PRS (either as categorical, $p = 0.4$, or continuous, $p = 0.8$) or deviation from the proportional hazards assumption ($p > 0.05$ for all covariables in models with categorical and continuous PRS, using Schoenfeld residuals).

**Sensitivity analysis addressing missingness.** Results of a sensitivity analysis based on multiply imputed data are shown in Supplementary Table 7. The results were similar to the primary analysis, though the adjusted hazard ratio for the high risk PRS group was higher (4.46, 95% CI 3.52, 5.66).

### Estimation of AAA prevalence
In a Fine and Gray competing risks model including only sex, the modelled estimated prevalence of AAA in men at age 65 was 0.41%. This finding suggests our data identified just under half of the AAAs observed in NAAASP at this age, where the observed prevalence was around 0.91%[21]. The lower yield of AAA cases in UK Biobank was likely due to a combination of the "healthy cohort" effect and the outcome definition which only captured AAAs identified at a hospital visit. We derived a scaling factor (F) to estimate population PRS-specific AAA prevalences for the DES from the UK Biobank AAA prevalence modelled here as the ratio of the NAAASP and UK Biobank prevalences in men (0.91/0.41 = 2.2). No equivalent data were available for UK women due to the lack of a systematic screening programme, so in our modelling we assumed F to be the same for both men and women.

### Discrete event simulation modelling
**Incremental net benefit by PRS and smoking status.** Figures 2 and 3 show the impact on incremental net monetary benefit (INB) for a range of different ages at invitation in men (followed up from age 60) and women (followed up from age 65) respectively, based on a willingness-to-pay of £30,000 per quality-adjusted life-year (QALY). The results depend on AAA prevalence at the starting age, so the INB is presented for a range of prevalences (modelled as known input parameters) corresponding to different sub-groups based on PRS risk and smoking status, i.e. these are sub-group INB estimates. Vertical lines indicated on Figs. 2 and 3 represent point estimates of smoking/PRS sub-group prevalences, though these are estimated with uncertainty. Results

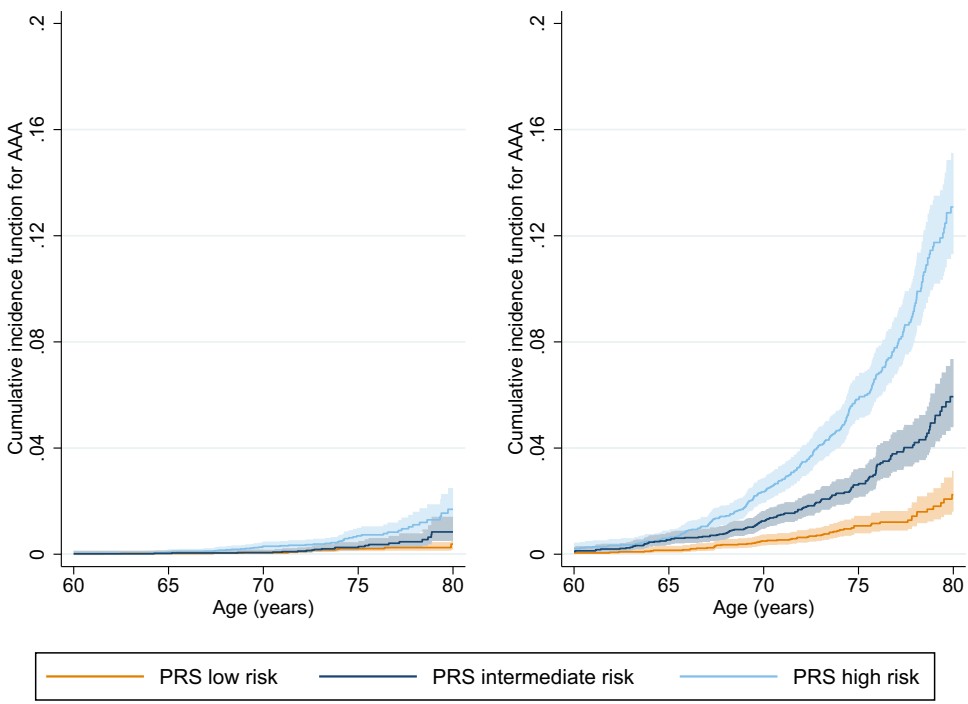

**Fig. 1 | Observed non-parametric cumulative incidence curves for recorded AAA in the UKB test set.** The CIF is shown separately for women (left) and men (right). PRS groups correspond to tertiles of PRS risk. Shaded areas represent 95% confidence intervals for the cumulative incidence function. AAA abdominal aortic aneurysm, UKB UK Biobank, PRS polygenic risk score.

based on PRS risk or smoking status alone, and based on a willingness-to-pay of £20,000 per QALY, are given in Supplementary Figs. 2–5.

For men, when AAA prevalence was below 0.2% at age 60, the INB was negative at all invitation ages - indicating no benefit in offering screening (Fig. 2). Ever smokers with low/intermediate PRS and ex-smokers with low PRS had AAA prevalences estimated in this range. For current smokers in the highest PRS risk group (prevalence around 1.5% at age 60), there was a positive INB at all ages at invitation for this subgroup of the population, and an increase in INB for invitation earlier than age 65; the INB was maximised by inviting this group at age 60. For all the remaining smoking/PRS subgroups (i.e. never smokers with high PRS, ex-smokers with intermediate/high PRS and current smokers with low/intermediate PRS; prevalences range from 0.2–0.7% at age 60), there was evidence that invitation to screening confers a positive net benefit, particularly at earlier invitation ages. The results showed the highest INB in these subgroups occurred following invitation between ages 60 and 62. There was little difference in terms of INB between inviting at age 60 or 62 because of the trade-off between missing late-developing AAAs and the relatively small numbers who may benefit from early intervention when the prevalence at age 60 is below 1%.

In women, when AAA prevalence was below 0.25% at age 65, the sub-group INB was negative at all invitation ages (Fig. 3). When considering subgroups of the population in isolation, only current smokers with intermediate PRS (prevalence 0.35% at age 65) and current smokers with high PRS (prevalence 0.8% at age 65) showed a small positive benefit. For the former this occurred at invitation age 70, and for the latter at invitation ages 65 and 70.

**Incremental net benefit by population strategy.** Evaluation of the impact of strategies stratified by PRS and/or smoking scaled to the whole population are given in Tables 2 and 3, i.e. these are population-level INB estimates.

In men, for a healthcare provider willing to pay £30,000 per QALY gained, offering universal screening at age 62 improved the incremental net benefit compared to the current strategy of universal screening at age 65 (mean population INB £41v £11, Table 2). When extrapolated to the population of 348,000 60-year-old men in England[22], this equated to an overall net gain of around £10.4 m at a willingness-to-pay of £30,000 per QALY. Using PRS-specific or smoking-specific age at invitation improved this further (both mean population INB £42; net gain £10.9 m over the whole population). The largest gains in men arose from a policy stratifying on a combination of both PRS and smoking, with an estimated net gain of £12.6 m over the whole population. Specifically, this policy invites men who are current smokers with high PRS risk at age 60, no invitation for never-smokers with low/intermediate PRS or ex-smokers with low PRS, and invites the remainder of the male population at age 62. In addition, this policy demonstrated the largest reduction in a number of scans, with a 41% reduction compared to the current policy (Table 2). In comparison, a policy of inviting all at age 62 marginally increased scans compared to the current policy.

In women, adopting a policy of inviting current smokers with high and intermediate PRS at ages 65 and 70 respectively conferred a modest improvement over the current approach of no screening (population INB around £3, Table 3). This equates to around £0.9 m over the population of 298,000 65-year-old women in England[22]. This is marginally higher than the overall gain over the population of £0.6 m estimated for a policy of inviting all current female smokers at age 65. Inviting all women with high PRS at age 70 conferred a negative population INB.

## Discussion

Our results may have several implications. First, it highlights that leveraging shared genetic aetiology by combining information from multiple traits can substantially improve PRS performance. Our most predictive PRS not only outperformed all other models without this information, but it also significantly outperformed two recent PRS by Roychowdhury et al.[23] and Wang et al.[24], whose AUCs were 0.693 and 0.608, respectively, versus our own best model's 0.708

**Table 1 | Hazard ratios for recorded AAA from multivariable Cox regression**

| Risk factor | HR (95% CI) | p-value[a] |
|---|---|---|
| PRS group | | <0.001 |
| Low risk | 1 | |
| Intermediate risk | 2.13 (1.61, 2.82) | |
| High risk | 3.70 (2.86, 4.80) | |
| Sex | | <0.001 |
| Female | 1 | |
| Male | 4.56 (3.54, 5.88) | |
| Townsend deprivation index (per 1 unit increase) | 1.02 (0.99, 1.05) | 0.1 |
| Alcohol intake | | 0.001 |
| Non-drinker | 1 | |
| Drinker | 0.57 (0.42, 0.79) | |
| Family history of CVD | | 0.6 |
| No | 1 | |
| Yes | 1.05 (0.88, 1.25) | |
| Diabetic | | 0.6 |
| No | 1 | |
| Yes | 1.09 (0.78, 1.53) | |
| Smoking status | | <0.001 |
| Never smoker | 1 | |
| Ex-smoker | 2.36 (1.82, 3.05) | |
| Current smoker | 7.74 (5.83, 10.29) | |
| BMI | | 0.2 |
| (per kg/m² increase) | 1.02 (0.99, 1.04) | |
| Systolic blood pressure | | 0.3 |
| (per 10 mm/Hg increase) | 0.97 (0.93, 1.02) | |
| Anti-hypertensive medication | | <0.001 |
| No | 1 | |
| Yes | 2.88 (2.36, 3.52) | |
| Total cholesterol | | 0.02 |
| (per mmol/L increase) | 1.12 (1.01, 1.23) | |
| HDL cholesterol | | <0.001 |
| (per mmol/L increase) | 0.27 (0.20, 0.39) | |
| Lipid-lowering medication | | <0.001 |
| No | 1 | |
| Yes | 2.82 (2.27, 3.51) | |

*HR* hazard ratio, *CI* confidence interval, *PRS* polygenic risk score, *CVD* cardiovascular disease, *BMI* body mass index, *HDL* high density lipoprotein.

[a]two-sided Wald test without correction for multiple comparisons for continuous and dichotomous variables; two-sided likelihood-ratio test without correction for multiple comparisons for categorical variables.

(Supplementary Table 3). This is particularly relevant for lower prevalence diseases such as AAA, where there are fewer GWAS studies with a lower number of cases (compared to coronary artery disease, for example). Second, current screening may not be optimal in terms of cost-effectiveness and does not capture AAAs in some individuals before rupture occurs. PRS could be used to identify individuals in the population who are currently not eligible for AAA screening, but are at high risk (such as females who are smokers or with high PRS, or men aged less than 65 who are smokers or with high PRS), and to optimise the timing of invitation. We provide evidence from a previously-validated simulation model that PRS-stratified AAA screening has the potential to improve cost-effectiveness over current strategies.

Results from the simulation model suggested that adopting a stratified approach to screening invitation age in men, and applying targeted screening in women, could be cost-effective for a healthcare provider willing to pay £30,000 per QALY when compared to the current strategy. In men, whilst PRS-based stratification conferred some increase in incremental net benefit, improvement arising from smoking-based stratification was almost identical. However, the largest gains were estimated to arise from using PRS and smoking in combination. Our modelling suggests that inviting male current smokers with high PRS at age 60 alongside no invitation for those with very low risk (never/ex-smokers with low PRS and never smokers with intermediate PRS) and invitation at age 62 for the remaining men, may improve cost-effectiveness. In women, combining smoking and PRS information enabled identification of a subgroup (current smokers with high and intermediate PRS, invited at age 65 and 70 respectively) in whom screening was cost-effective.

Our proposed smoking and PRS-risk-based AAA screening invitation strategy incorporates earlier invitations than currently offered for those in the highest PRS-risk groups where AAA may develop earlier. Conversely, those without a high PRS are offered a later (or no) invitation. This does not necessarily imply an increase in missed AAAs for these individuals, since there is a trade-off in optimising identification of AAAs (i.e. early enough to maximise capture before rupture, late enough to minimise missing AAAs that develop post-screening) with maximising life-years gained by intervening at younger ages. In addition, the continued exclusion of very low risk groups from screening (women who are never/ex-smokers and those with low PRS) limits the potential for harms associated with overdiagnosis in those who would likely not go on to experience AAA intervention or rupture[19].

We report a strong association between our AAA PRS and reported AAA outcome alongside an apparently more modest improvement for a PRS-stratified screening approach in terms of mean INB. This is in part because benefits are only accrued in those with an AAA, but are averaged over the whole screened population when calculating cost-effectiveness measures. Additionally, the possibility for a PRS that is strongly associated with the outcome to translate into a modest impact on a screening programme has been previously documented[25]. For a given PRS cut-off, even though the proportion of cases to non-cases may be considerably lower amongst those with low PRS (reflecting a large hazard or odds ratio when assessing association), the majority of cases may still occur in this group when a relatively small proportion of the population has a high PRS. If translated into a screening programme based on a PRS cut-off, this would result in missing many cases. We address this here by taking a lifetime perspective and by considering variations to the timing of screening invitation in addition to comparisons to no screening. The use of the DES to evaluate the outcomes associated with each policy ensures that increases in AAA prevalence and AAA ruptures during the unscreened period for those with low/intermediate PRS are accounted for in the results.

## Limitations

Our work makes the assumption that generating a PRS profile might, in the future, not carry an additional cost, given expectations that PRS might one day be offered at the population-level and utilised across a range of diseases, implying a negligible per-trait cost. We acknowledge this is a major assumption. However, this scenario - i.e. one in which genomic information will become part of routine healthcare and available for screening purposes[26–30] - has been advocated by some leading authorities[31]. Such a programme of systematic collection of genomics data - with a reasonable level of uptake in the population - would, of course, need to be in place to facilitate the implementation of a PRS-stratified screening strategy as described here. Furthermore, published estimates of genotyping are highly variable[28,29,32–34]. Applying the lowest of these figures (USD $29, £23), as expected, reduces the net gain over population considerably, to £2.9 m for the PRS-based strategy and £4.6 m for the combined PRS and smoking strategy.

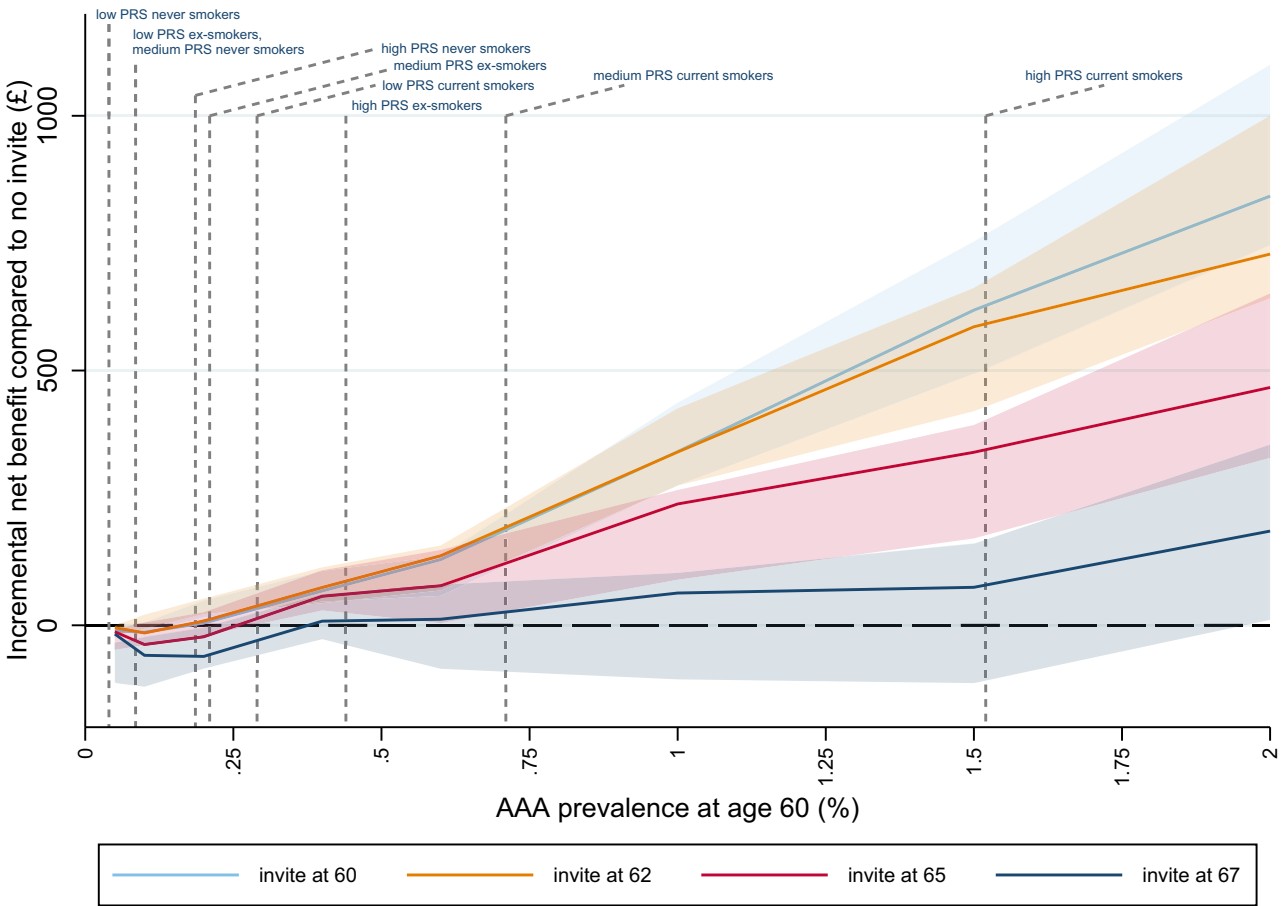

**Fig. 2 | Incremental net benefit compared to no invitation, by age at invitation and baseline prevalence at age 60 in men.** INB is evaluated at a willingness-to-pay of £30,000 per QALY, based on 1 M hypothetical individuals in the DES. Points plotted are point estimates. Shaded areas represent 95% uncertainty intervals derived from 100 bootstrap PSA samples. PRS/smoking sub-group prevalences estimated from UKB test set as CIF x inflation factor; indicated on the x-axis. INB incremental net benefit, QALY quality-adjusted life-year, DES discrete event simulation, PSA probabilistic sensitivity analysis, PRS polygenic risk score, UKB UK Biobank, CIF cumulative incidence function, AAA abdominal aortic aneurysm.

Genotyping costs up to £31 per test are estimated to result in a positive net gain over the population compared to the current strategy, though gains are lower than the smoking-only strategy.

We modelled AAA incidence data in UK Biobank, which is not directly linked to the AAA national screening programme in the UK. UK Biobank also has a "healthy volunteer bias" that could impact the performance of our PRS – however, we anticipate that the cost-effectiveness of PRS-stratified AAA screening would be more favourable, given the expected higher and earlier disease prevalence in a cohort without healthy volunteer bias. Relatedly, we do not present a risk prediction model or provide optimism-corrected model evaluations. Our test set, while it did not overlap with our training set, was from the same population (UK Biobank). Before wider deployment, our models may need to be evaluated in an external validation set. The PRS contains only common genetic variants (MAF > 0.01) and there may be rare and low-frequency variants that impact risk of AAA. We assume that the outcomes of AAA repair, growth rates/rupture risk of AAA, and non-AAA mortality rates are consistent across all strata of smoking and PRS for AAA susceptibility, and by age, which may not be the case. We have modelled upon an assumption that there is one single screening point, but alternative approaches with a repeat screening of high-risk individuals may further improve clinical and cost-effectiveness outcomes. In addition, alternative stratified approaches incorporating clinical risk factors such as BMI and blood pressure may further improve cost-effectiveness but have not been evaluated here. Finally, due to the limited data availability, our study

participants were restricted to individuals of European ancestry. The portability of PRS across populations has been shown to be reduced due to differences in patterns of linkage disequilibrium, allele frequencies and effect sizes[35], thus we expect that our model to be mainly relevant to European ancestry populations. Due to the low prevalence of AAA, large population studies across multiple ancestries will be required, such as those provided by the *Our Future Health*[36] and *All of Us*[37] projects, to increase the representation of more diverse populations in future AAA studies.

In summary, we have developed a novel PRS for AAA that demonstrates an independent association with incident AAA above clinical risk factors. We found that PRS-informed screening could identify subpopulations who are currently excluded from screening policy (such as intermediate and high PRS female smokers), in whom screening may be cost-effective. We also report that screening in men could be optimised by varying the age at invitation according to polygenic risk and smoking status.

## Methods
### Development of a polygenic risk score
The polygenic risk score presented here relies on two AAA data sources as follows: (i) GWAS performed in the UK BioBank (UKB)[38] (1068 cases and 127,011 controls, selected in a way to exclude all individuals who are part of the AAAGen study), and (ii) summary level data from the AAAgen cohort[8] without the UKB (effective sample size of ~104,179; see details in Supplementary Methods Section 1). The full study details

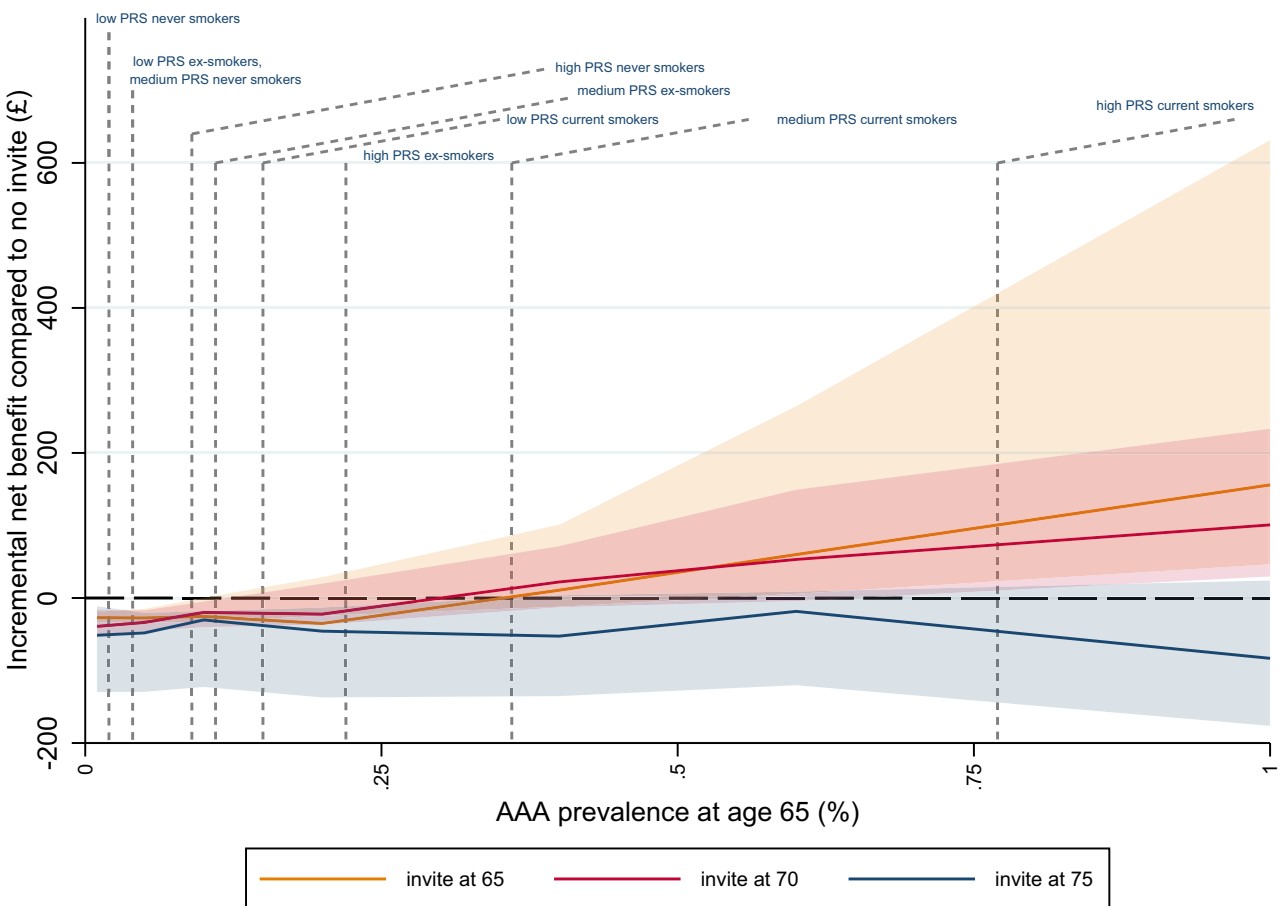

**Fig. 3 | Incremental net benefit compared to no invitation, by age at invitation and baseline prevalence at age 65 in women.** INB is evaluated at a willingness-to-pay of £30,000 per QALY, based on 1 M hypothetical individuals in the DES. Points plotted are point estimates. Shaded areas represent 95% uncertainty intervals derived from 100 bootstrap PSA samples. PRS/smoking sub-group prevalences estimated from UKB test set as CIF x inflation factor; indicated on the x-axis. INB incremental net benefit, QALY quality-adjusted life-year; DES discrete event simulation, PSA probabilistic sensitivity analysis, PRS polygenic risk score, UKB UK Biobank, CIF cumulative incidence function, AAA abdominal aortic aneurysm.

can be found in Supplementary Table 1. To further improve predictive performance, we also considered summary association data from other phenotypes which we believed may share genetic aetiology with AAA: (i) coronary heart disease (CHD)[8], (ii) stroke[39], and (iii) conditions related to AAA in the UKB (Supplementary Table 2). We adopt the assumption of the GWAS from which our PRS was sourced from, i.e. that the risk liability can be estimated from the combined set of pre-valent and incident cases. Detailed data processing, quality control steps applied to each dataset, and the full list of ICD10 codes for the "AAA-related" phenotype can be found in the Supplementary Methods.

To maximise performance, we performed a fixed-effect meta-analysis on all AAA studies. We then evaluated two PRS pre-processing methods that exploit genetic overlaps between the aetiologies of dif-ferent traits, shaPRS[16], and MTAG[40], to additionally integrate infor-mation from CAD, stroke, and AAA-related traits. To generate the final PRS from the subset of the 831,447 SNPs that met our quality control criteria in the HapMap3 panel, we also evaluated two methods, PRS-CS and LDpred2, both via their 'auto' options. All PRS were evaluated for calibration via a Hosmer-Lemeshow test modified for larger sample sizes[41] and the best individual PRS was chosen by comparing the AUC (from a univariable logistic regression model assuming a linear rela-tionship with continuous PRS) and the squared Pearson correlation coefficient ($r^2$) between predicted and observed phenotypes evaluat-ing the performance of all PRS models in a non-overlapping, randomly selected test set of 91,731 European ancestry individuals in UKB, which included 869 (prevalent and incident combined) AAA cases.

### Modelling the independent association of PRS with AAA

The non-overlapping subset of 91,731 individuals in UK Biobank also served as the test set for our risk prediction model, using 740 incident time-to-AAA as the outcome of interest to allow for appropriate adjustment of other potentially time-varying risk factors. Date and cause of death together with the date of loss-to-follow-up were col-lected. The outcome was defined as first hospital inpatient admission with AAA code (either AAA-related surgical procedure or AAA code recorded at admission for other cause) or AAA death. This includes ICD-10 codes I71.3 and I71.4 together with surgical procedure codes (see Supplementary Table 4 for full list); as such this definition does not relate to specific diagnostic criteria such as aneurysm size. This outcome may strictly be defined as "recorded AAA" to reflect the fact that it captures only a proportion of existing AAAs in the population (since those with an unidentified unruptured AAA and those with an unruptured AAA who do not attend hospital in the follow-up period will not be included here), though for simplicity will be hereafter referred to as AAA. Under this definition, the AAA may be known either because of rupture, surgical intervention or detection either via NAAASP or opportunistically. Individuals with a prevalent AAA event at entry into UKB were excluded from this analysis.

Cox regression models on the age time-scale were used to explore the independent association between PRS and time to AAA with and without adjustment for known risk factors. Individuals were censored on the date of loss-to-follow-up, end-of-follow-up on 31st March 2021, age 80, or date of non-AAA mortality, whichever was sooner. Models

**Table 2 | Population incremental net benefit estimates arising from different invitation age strategies in men**

| | Strategy | | | | |
|---|---|---|---|---|---|
| | Current: Invite all age 65 | Invite all age 62 | Invite age determined by PRS[a] | Invite age determined by smoking[b] | Invite age determined by PRS + smoking[c] |
| Mean INB (95% UI)[d] | £11.03 (-£1.74, £37.12) | £40.89 (£25.75, £58.49) | £42.49 (£25.86, £54.63) | £42.31 (£36.78, £66.66) | £47.28 (£39.29, £70.87) |
| Mean QALYs | 17.1603 | 17.1614 | 17.1611 | 17.1611 | 17.1612 |
| Mean costs | £271.79 | £275.63 | £265.96 | £265.91 | £263.80 |
| Net gain over population[e] | NA | £10.4 m | £10.9 m | £10.9 m | £12.6 m |
| *Events per 10,000 invited* | | | | | |
| Scans[f] | 9229 | 9560 | 6250 | 6196 | 5474 |
| Elective operations | 108.3 | 111.2 | 110.9 | 110.8 | 111.0 |
| Emergency operations | 42.9 | 42.1 | 42.0 | 42.2 | 41.9 |
| Ruptures | 118.1 | 115.9 | 116.1 | 116.3 | 115.9 |
| AAA-related deaths | 97.3 | 96.0 | 96.2 | 96.3 | 96.2 |

PRS polygenic risk score, INB incremental net benefit, QALY quality-adjusted life-year, AAA abdominal aortic aneurysm, DES discrete event simulation.
Note: These results account for the differing proportions of the population within each sub-group, as observed in the test set.
[a]Invite intermediate and high PRS only, at age 62 (maximises PRS-specific INBs).
[b]Invite ex and current smokers only, at age 62 (maximises smoking-specific INBs).
[c]Invite current smokers with high PRS at age 60; no invite for never smokers with low/intermediate PRS; no invite for ex-smokers with low PRS; invite remainder at age 62 (maximises PRS/smoking-subgroup-specific INBs).
[d]Mean incremental net benefit compared to no invite derived from point estimate; 95% uncertainty interval derived 5th/95th centiles of INB from 100 PSA bootstrap samples.
[e]based on UK population estimates for 60-year-old men (N = 348,000). Comparison to current strategy, based on willingness-to-pay of £30,000 per QALY.
[f]includes both initial screen and follow-up monitoring scans for those with detected AAA; input parameter for attendance set to 75% for men / 72% for women (as per original DES).

**Table 3 | Population incremental net benefit estimates arising from different invitation age strategies in women**

| | Strategy | | | |
|---|---|---|---|---|
| | Current: no invite | Invite high PRS only[a] | Invite current smokers only[b] | Invite age determined by PRS + smoking[c] |
| Mean INB (95% UI)[d] | N/A | -£3.74 (-£8.25, £6.24) | £2.09 (-£0.06, £14.01) | £2.95 (£0.47, £12.78) |
| Mean QALYs | 15.2443 | 15.2445 | 15.2445 | 15.2445 |
| Mean costs | £72.57 | £81.63 | £75.14 | £74.51 |
| Net gain over population[e] | NA | -£1.1 m | £0.6 m | £0.9 m |
| *Events per 10,000 invited* | | | | |
| Scans[f] | 0 | 2629 | 569 | 395 |
| Elective operations | 17.5 | 18.1 | 17.9 | 17.8 |
| Emergency operations | 18.7 | 18.5 | 18.6 | 18.6 |
| Ruptures | 72.4 | 71.8 | 72.0 | 72.1 |
| AAA-related deaths | 65.5 | 65.2 | 65.3 | 65.3 |

PRS polygenic risk score, INB incremental net benefit, QALY quality-adjusted life-year, AAA abdominal aortic aneurysm, DES discrete event simulation.
Note: These results account for the differing proportions of the population within each sub-group, as observed in the test set.
[a]Invite high PRS at age 70; otherwise no invite.
[b]Invite current smokers at age 65; otherwise no invite.
[c]Invite current smokers with high PRS at age 65; invite current smokers with intermediate PRS at age 70; otherwise no invite (maximises PRS-smoking-subgroup-specific INBs).
[d]Mean incremental net benefit compared to no invite derived from point estimate; 95% uncertainty interval derived 5th/95th centiles of INB from 100 PSA bootstrap samples.
[e]based on UK population estimates for 65-year-old women (N = 298,000). Comparison to current strategy, based on willingness-to-pay of £30,000 per QALY.
[f]includes both initial screen and follow-up monitoring scans for those with detected AAA.

were fitted using data for men and women combined, with adjustment for sex and testing for sex x PRS interaction, due to the low number of AAA cases in women. We selected conventional risk factors based on their likely availability at the clinic: date of birth, BMI, self-reported smoking status (never, ex, current), self-reported alcohol consumption (non-drinker, drinker), diabetes, anti-hypertensive drug use, lipid-lowering drug use, systolic blood pressure (SBP), high-density lipoprotein (HDL), total cholesterol, Townsend deprivation index, and family history of CVD, as recorded at entry into UKB. Linearity was assumed for continuous variables in the Cox modelling. In contrast to the PRS development stage where exploration of the associations between genetic factors and blood pressure implicates the use of underlying blood pressure and cholesterol (i.e. adjusted for medication to estimate the pre-medication values), this stage of modelling uses baseline SBP and cholesterol measurements as the relevant predictors of future AAA risk.

Tertiles of PRS were used to allow continuity with the simulation modelling component of this work, which requires grouping of PRS for evaluation of stratified screening. Tertiles were selected to minimise impact on precision of estimates - particularly important in the analysis of this relatively rare condition - and for ease of interpretation. Key results are also presented per standard deviation increase in PRS. In a sensitivity analysis, we additionally explore the impact of using multiple imputations to account for missing values of risk factors. Full

details of methods are given in the Supplementary Information. All post-PRS-development statistical analysis was conducted using Stata v17.0 (StataCorp, College Station, Texas 77845, USA).

## Discrete event simulation modelling

We further use the UK Biobank data to explore the potential impact of setting PRS-sex-specific ages for screening invitation, and determine the costs and benefits of this AAA screening programme. Fine and Grey regression modelling is employed to calculate cumulative incidence functions (CIFs) for time to AAA whilst accounting for competing risks due to non-AAA mortality. This model is used to estimate the prevalence of AAA in men at age 65 in UKB, which is benchmarked against the observed prevalence of AAA at this age in the National Health Service Abdominal Aortic Aneurysm Screening Programme (NAAASP). This measure gives an indication of the proportion of AAAs that are captured by the definition of recorded AAA used in UKB and is used as a scaling factor (F) for sub-group baseline prevalences estimated from the Fine and Gray model at age 60 in men and age 65 in women that are then taken forward to make group-specific inferences on results from the DES. Since women are not screened in the UK, it is not possible to estimate a separate scaling factor for women, so we assume that the ratio of recorded:underlying AAAs (and thus, F) is the same in women as in men.

To explore how a PRS-sex-specific age at invitation strategy may influence long-term outcomes, we adapted a previously developed discrete event simulation (DES) model for AAA screening, run in R Statistical Software v4.2.1 (R Foundation for Statistical Computing; http://www.r-project.org). The SWAN model[19] This DES combines growth and rupture rate models reflecting the natural history of AAA with information from screening programmes on uptake, detection (including that taking place outside of systematic screening), distribution of AAA diameters, and both elective and emergency surgical intervention. The model input parameters are informed by systematic review and meta-analysis where possible, with non-AAA mortality informed using national summary statistics.

We use the model to track clinical events and costs from age 60 to 95 (men) or age 65 to 95 (women) in 1 m hypothetical individuals, separately for men and women. Younger ages are not considered here since model input parameters (e.g. attendance at screening, rates of dropout, incidental detection and re-intervention) are largely derived from populations aged 65 + . Models are run for a range of prevalences at the starting age (60 in men; 65 in women) and for different invitation ages between 60 and 67 in men and between 65 and 75 in women, treating the prevalences as known input parameters and plotting final results over the full range of values.

Results are summarised in terms of mean quality-adjusted life-years (QALYs) and mean costs, for each baseline prevalence and invitation age. The incremental net benefit (INB) is also calculated to provide an estimate of the mean net monetary gain for the invitation at a particular age compared to a no-invite strategy (for a given baseline prevalence) by assigning a willingness-to-pay (WTP) per QALY gained, set here at £30,000 per QALY. Population strategy is optimised by identifying the age at invitation corresponding to the highest INB within each sub-group. Results corresponding to each sub-group are then combined to provide an overall population cost-effectiveness by scaling according to the observed proportion in the UKB test set.

## Reporting summary

Further information on research design is available in the Nature Portfolio Reporting Summary linked to this article.

## Data availability

Two datasets are used in the analyses presented in this paper: (1) AAAgen (PRS development), and (2) UK Biobank (PRS development, modelling the independent association of PRS with AAA, and DES modelling). AAAgen meta-analysis summary statistics are freely available to download from https://csg.sph.umich.edu/willer/public/AAAgen2023/. UK Biobank genotype and phenotype data can be accessed via the UKB research analysis platform (RAP): https://ukbiobank.dnanexus.com/landing. The Research Analysis Platform is open to researchers who are listed as collaborators on UKB-approved access applications.

## Code availability

Code to perform all PRS development analyses reported in this manuscript is available at https://www.github.com/mkelcb/aaa-paper. Our final PRS model is deposited in the PGS Catalog[42] at https://www.pgscatalog.org/ and is available to download under accession code PGS003429. The full DES model is available at https://github.com/mikesweeting/AAA_DES_model.

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

## Acknowledgements

L.G.K., M.I., L.P., J.D., A.W. and E.D.A were supported by core funding from the British Heart Foundation (RG/18/13/33946: RG/F/23/110103), NIHR Cambridge Biomedical Research Centre (NIHR203312) [*], BHF Chair Award (CH/12/2/29428), Cambridge BHF Centre of Research Excellence (RE/18/1/34212; RE/24/130011), and by Health Data Research UK, which is funded by the UK Medical Research Council, Engineering and Physical Sciences Research Council, Economic and Social Research Council, Department of Health and Social Care (England), Chief Scientist Office of the Scottish Government Health and Social Care Directorates, Health and Social Care Research and Development Division (Welsh Government), Public Health Agency (Northern Ireland), British Heart Foundation and the Wellcome Trust. J.D. holds a British Heart Foundation Professorship and a NIHR Senior Investigator Award [*]. L.G.K., E.D.A. and A.M.W. were supported by the National Institute for Health and Care Research (NIHR) Blood and Transplant Research Unit (BTRU) in Donor Health and Behaviour (NIHR203337). M.K. is funded by the BHF Cambridge CRE (RE/18/1/34212). A.M.W is part of the BigData@Heart Consortium, funded by the Innovative Medicines Initiative-2 Joint Undertaking under grant agreement No 116074. A.M.W. was also supported by the BHF-Turing Cardiovascular Data Science Award (BCDSA\100005). M.I. was also supported by the UK Economic and Social Research 878 Council (ES/T013192/1). This research used data assets made available by National Safe Haven as part of the Data and Connectivity National Core Study, led by Health Data Research UK in partnership with the Office for National Statistics and funded by UK Research and Innovation (research which commenced between 1st October 2020 – 31st March 2021 grant ref MC_PC_20029; 1st April 2021 –30th September 2022 grant ref MC_PC_20058). This research has been conducted using the UK Biobank Resource under Application Number 7439 (https://www.ukbiobank.ac.uk). For the purpose of open access, the author has applied a Creative Commons Attribution (CC BY) licence to any Author Accepted Manuscript version arising from this submission. *The views expressed are those of the authors and not necessarily those of the NIHR or the Department of Health and Social Care.

## Author contributions

M.K. and S. H. conceived and designed the study. S.H. and L.G.K. managed the project. M.K. and L.G.K. wrote the manuscript with critical input from S. H. M.K. performed QC in the UKB and developed the PRS. L.G.K. conducted the health-economic modelling and DES simulations. M.K., J.D., E.D.A., M.I., J.O'S., L.P., T.R., M.J.S., A.M.W., S.H., and L.G.K. contributed feedback and suggestions to the manuscript and approved the submitted version.

## Competing interests

J.D. serves on scientific advisory boards for AstraZeneca, Novartis, and UK Biobank, and has received multiple grants from academic, charitable and industry sources outside of the submitted work. M.I. is a trustee of the Public Health Genomics (PHG) Foundation, a member of the Scientific Advisory Board of Open Targets, and has research collaborations

with AstraZeneca, Nightingale Health and Pfizer which are unrelated to this study. M.J.S. reports full-time employment with AstraZeneca and AstraZeneca stock ownership. S.H. is an employee at Genomics PLC. The other authors declare no competing interests.
