## [Peer Review File · Nature Communications]

Evaluating the cost-effectiveness of polygenic risk score-stratified screening for abdominal aortic aneurysmEditorial Note: This manuscript has been previously reviewed at another journal that is not operating a transparent peer review scheme. This document only contains reviewer comments and rebuttal letters for versions considered at *Nature Communications*.

REVIEWERS' COMMENTS

Reviewer #2 (Remarks to the Author):

In their revised manuscript, the authors have largely clarified my questions, and have improved their description of PRS development/assessment. I still have a few additional points:

- The number of individuals in the test set should be clarified – 921 in the methods, 869 in Supplementary Figure 1; more broadly, Supplementary Figure 1 should be modified to better correspond to what is listed in the Methods (eg. “Modeling the independent association of PRS with AAA” focuses on 740 incident events (vs. 921 listed earlier in Methods, 869 in figure).
- The authors focus on a “simple strategy based on clinically available risk factors.” However, age/sex/smoking are likely more limited than realistic/clinically-actionable risk models might otherwise implement. For example, risk assessments for coronary artery disease including consideration of lipid levels, presence of diabetes, blood pressure, medications, etc. As referenced in my prior comments, PMID: 36378208 considered a relatively simple set of clinical variables that should be available for most individuals. While I appreciate the author’s interest in comparing to the current baseline, identifying the most effective models (at lowest cost) is ultimately what will drive greatest clinical impact. Each strategy has tradeoffs in terms of ease/feasibility/cost/etc. I think that the authors should acknowledge that their proposed PRS-based screening strategy has not been compared to models including additional simple variables like height/weight/blood pressure. The value of additional clinical variables will need to be assessed in future work to identify the optimal screening strategy. The current work is important, demonstrating that PRS may offer one opportunity to improve upon the current baseline. Whether this is actually the best/most effective path to pursue remains uncertain.

Reviewer #2 (Remarks on code availability):

The github repository contains a descriptive README file. I have not installed the code or tried to reproduce the analyses.

Reviewer #3 (Remarks to the Author):

Kelemen and colleagues present work in which they generated a novel PRS to predict AAA leveraging shared genetic effects across correlated traits. Following, they evaluated cost-effectiveness of screening informed by genetic risk.

Overall, the authors appropriately responded to reviewer comments.

1. A primary finding is that the best performing PRS leveraged CAD + AAA-related traits. It would be helpful to highlight this result in the abstract. As it currently reads, it is unclear in the abstract that the multi-trait PRS outperformed other methods.
2. Similarly, it would be helpful to include some additional clarifying information in the results section. I might consider listing some of the AAA-related traits in the main text of the manuscript (line 89-90).
3. I might also consider including in the text a direct comparison of the HR per 1-SD PRS (line 110-111) between the multi-trait PRS and AAA PRS in the main text of the manuscript.
4. minor: there appear to be two instances of supplementary table 3 in the text.

Response to reviewers:

Reviewer #2 (Remarks to the Author):

In their revised manuscript, the authors have largely clarified my questions, and have improved their description of PRS development/assessment. I still have a few additional points:

- The number of individuals in the test set should be clarified – 921 in the methods, 869 in Supplementary Figure 1; more broadly, Supplementary Figure 1 should be modified to better correspond to what is listed in the Methods (eg. “Modeling the independent association of PRS with AAA” focuses on 740 incident events (vs. 921 listed earlier in Methods, 869 in figure).

We thank the reviewer for drawing our attention to this error and lack of clarity. The number 921 referred to a pre-QC subset, which was not actually used for our models. So the 921 has now been corrected to 869. We had 869 cases in total, which included both prevalent and 740 incident cases. We have now clarified throughout the number used for each purpose (869 for the PRS development based on prevalent and incident cases combined, and 740 for the modelling based on incident events only).

- The authors focus on a “simple strategy based on clinically available risk factors.” However, age/sex/smoking are likely more limited than realistic/clinically-actionable risk models might otherwise implement. For example, risk assessments for coronary artery disease including consideration of lipid levels, presence of diabetes, blood pressure, medications, etc. As referenced in my prior comments, PMID: 36378208 considered a relatively simple set of clinical variables that should be available for most individuals. While I appreciate the author’s interest in comparing to the current baseline, identifying the most effective models (at lowest cost) is ultimately what will drive greatest clinical impact. Each strategy has tradeoffs in terms of ease/feasibility/cost/etc. I think that the authors should acknowledge that their proposed PRS-based screening strategy has not been compared to models including additional simple variables like height/weight/blood pressure. The value of additional clinical variables will need to be assessed in future work to identify the optimal screening strategy. The current work is important, demonstrating that PRS may offer one opportunity to improve upon the current baseline. Whether this is actually the best/most effective path to pursue remains uncertain.

Thank you for highlighting this important addition for our discussion. We have now added a sentence reflecting the possibility of further benefits through the incorporation of clinical risk factors in our limitations section: “In addition, alternative stratified approaches incorporating clinical risk factors such as BMI and blood pressure may further improve cost-effectiveness, but have not been evaluated here.”.

Reviewer #2 (Remarks on code availability):

The github repository contains a descriptive README file. I have not installed the code or tried to reproduce the analyses.

Reviewer #3 (Remarks to the Author):

Kelemen and colleagues present work in which they generated a novel PRS to predict AAA leveraging shared genetic effects across correlated traits. Following, they evaluated cost-effectiveness of screening informed by genetic risk.

Overall, the authors appropriately responded to reviewer comments.

1. A primary finding is that the best performing PRS leveraged CAD + AAA-related traits. It would be helpful to highlight this result in the abstract. As it currently reads, it is unclear in the abstract that the multi-trait PRS outperformed other methods.

This is indeed an important result, and we thank the reviewer for drawing our attention to this omission. We have now added requested information into the abstract:

“We find that leveraging related traits improves PRS performance (R^2) by 22.7%, relative to using information from AAA alone.”

2. Similarly, it would be helpful to include some additional clarifying information in the results section. I might consider listing some of the AAA-related traits in the main text of the manuscript (line 89-90).

We agree that this would improve the clarity of the text, so we specified the exact number (21) and have given 3 specific examples of the “AAA-related” phenotype:

“AAA-related is a composite phenotype of 21 conditions related to AAA (for example, Marfan's syndrome, myocardial infarction and hypertension, see Table S2 for a full list of conditions).”

3. I might also consider including in the text a direct comparison of the HR per 1-SD PRS (line 110-111) between the multi-trait PRS and AAA PRS in the main text of the manuscript.

We thank the reviewer for the useful suggestion to demonstrate the benefits of our approach via a numerical comparison between the final PRS that used additional traits versus the best PRS from AAA alone. However, as this is relevant for the PRS development, we added this additional comparison under the “AAA polygenic risk score development” section: *“Leveraging these additional related traits provided a performance advantage in R^2 of 22.7% over just using information from AAA alone (R^2 : 0.00530 v 0.00432).”*

4. minor: there appear to be two instances of supplementary table 3 in the text.

The second table was the old version from the first draft, which has now been removed.